# Regularizing and Optimizing LSTM Language Models

**Stephen Merity, Nitish Shirish Keskar & Richard Socher**
Salesforce Research
Palo Alto, CA 94301, USA
{smerity,nkeskar,rsocher}@salesforce.com

## Abstract

In this paper, we consider the specific problem of word-level language modeling and investigate strategies for regularizing and optimizing LSTM-based models. We propose the weight-dropped LSTM, which uses DropConnect on hidden-to-hidden weights, as a form of recurrent regularization. Further, we introduce NT-AvSGD, a non-monotonically triggered (NT) variant of the averaged stochastic gradient method (AvSGD), wherein the averaging trigger is determined using a NT condition as opposed to being tuned by the user. Using these and other regularization strategies, our AvSGD Weight-Dropped LSTM (AWD-LSTM) achieves state-of-the-art word level perplexities on two data sets: 57.3 on Penn Treebank and 65.8 on WikiText-2. In exploring the effectiveness of a neural cache in conjunction with our proposed model, we achieve an even lower state-of-the-art perplexity of 52.8 on Penn Treebank and 52.0 on WikiText-2. We also explore the viability of the proposed regularization and optimization strategies in the context of the quasi-recurrent neural network (QRNN) and demonstrate comparable performance to the AWD-LSTM counterpart. The code for reproducing the results is open sourced and is available at https://github.com/salesforce/awd-lstm-lm.

## 1 Introduction

Effective regularization techniques for deep learning have been the subject of much research in recent years. Given the over-parameterization of neural networks, generalization performance crucially relies on the ability to regularize the models sufficiently. Strategies such as dropout (Srivastava et al., 2014) and batch normalization (Ioffe & Szegedy, 2015) have found great success and are now ubiquitous in feed-forward and convolutional neural networks. Naïvely applying these approaches to the case of recurrent neural networks (RNNs) has not been highly successful however. Many recent works have hence been focused on the extension of these regularization strategies to RNNs; we briefly discuss some of them below.

A naïve application of dropout (Srivastava et al., 2014) to an RNN's hidden state is ineffective as it disrupts the RNN's ability to retain long term dependencies (Zaremba et al., 2014). Gal & Ghahramani (2016) propose overcoming this problem by retaining the same dropout mask across multiple time steps as opposed to sampling a new binary mask at each timestep. Another approach is to regularize the network through limiting updates to the RNN's hidden state. One such approach is taken by Semeniuta et al. (2016) wherein the authors drop updates to network units, specifically the input gates of the LSTM, in lieu of the units themselves. This is reminiscent of zoneout (Krueger et al., 2016) where updates to the hidden state may fail to occur for randomly selected neurons.

Instead of operating on the RNN's hidden states, one can regularize the network through restrictions on the recurrent matrices as well. This can be done either through restricting the capacity of the matrix (Arjovsky et al., 2016; Wisdom et al., 2016; Jing et al., 2016) or through element-wise interactions (Balduzzi & Ghifary, 2016; Bradbury et al., 2016; Seo et al., 2016).

Other forms of regularization explicitly act upon activations such as batch normalization (Ioffe & Szegedy, 2015), recurrent batch normalization (Cooijmans et al., 2016), and layer normalization (Ba

et al., 2016). These all introduce additional training parameters and can complicate the training process while increasing the sensitivity of the model.

In this work, we investigate a set of regularization strategies that are not only highly effective but which can also be used with no modification to existing LSTM implementations. The weight-dropped LSTM applies recurrent regularization through a DropConnect mask on the hidden-to-hidden recurrent weights. Other strategies include the use of randomized-length backpropagation through time (BPTT), embedding dropout, activation regularization (AR), and temporal activation regularization (TAR).

As no modifications are required of the LSTM implementation these regularization strategies are compatible with black box libraries, such as NVIDIA cuDNN, which can be many times faster than naïve LSTM implementations.

Effective methods for training deep recurrent networks have also been a topic of renewed interest. Once a model has been defined, the training algorithm used is required to not only find a good minimizer of the loss function but also converge to such a minimizer rapidly. The choice of the optimizer is even more important in the context of regularized models since such strategies, especially the use of dropout, can impede the training process. Stochastic gradient descent (SGD), and its variants such as Adam (Kingma & Ba, 2014) and RMSprop (Tieleman & Hinton, 2012) are amongst the most popular training methods. These methods iteratively reduce the training loss through scaled (stochastic) gradient steps. In particular, Adam has been found to be widely applicable despite requiring less tuning of its hyperparameters. In the context of word-level language modeling, past work has empirically found that SGD outperforms other methods in not only the final loss but also in the rate of convergence. This is in agreement with recent evidence pointing to the insufficiency of adaptive gradient methods (Wilson et al., 2017).

Given the success of SGD, especially within the language modeling domain, we investigate the use of averaged SGD (AvSGD) (Polyak & Juditsky, 1992) which is known to have superior theoretical guarantees. AvSGD carries out iterations similar to SGD, but instead of returning the last iterate as the solution, returns an average of the iterates past a certain, tuned, threshold $T$. This threshold $T$ is typically tuned and has a direct impact on the performance of the method. We propose a variant of AvSGD where $T$ is determined on the fly through a non-monotonic criterion and show that it achieves better training outcomes compared to SGD.

## 2 WEIGHT-DROPPED LSTM

We refer to the mathematical formulation of the LSTM,

$$
\begin{aligned}
i_t &= \sigma(W^i x_t + U^i h_{t-1}) \\
f_t &= \sigma(W^f x_t + U^f h_{t-1}) \\
o_t &= \sigma(W^o x_t + U^o h_{t-1}) \\
\tilde{c}_t &= \tanh(W^c x_t + U^c h_{t-1}) \\
c_t &= i_t \odot \tilde{c}_t + f_t \odot + \tilde{c}_{t-1} \\
h_t &= o_t \odot \tanh(c_t)
\end{aligned}
$$

where $[W^i, W^f, W^o, U^i, U^f, U^o]$ are weight matrices, $x_t$ is the vector input to the timestep $t$, $h_t$ is the current exposed hidden state, $c_t$ is the memory cell state, and $\odot$ is element-wise multiplication.

Preventing overfitting within the recurrent connections of an RNN has been an area of extensive research in language modeling. The majority of previous recurrent regularization techniques have acted on the hidden state vector $h_{t-1}$, most frequently introducing a dropout operation between timesteps, or performing dropout on the update to the memory state $c_t$. These modifications to a standard LSTM prevent the use of black box RNN implementations that may be many times faster due to low-level hardware-specific optimizations.

We propose the use of DropConnect (Wan et al., 2013) on the recurrent hidden to hidden weight matrices which does not require any modifications to an RNN's formulation. As the dropout operation is applied once to the weight matrices, before the forward and backward pass, the impact on

training speed is minimal and any standard RNN implementation can be used, including inflexible but highly optimized black box LSTM implementations such as NVIDIA's cuDNN LSTM.

By performing DropConnect on the hidden-to-hidden weight matrices $[U^i, U^f, U^o, U^c]$ within the LSTM, we can prevent overfitting from occurring on the recurrent connections of the LSTM. This regularization technique would also be applicable to preventing overfitting on the recurrent weight matrices of other RNN cells.

As the same weights are reused over multiple timesteps, the same individual dropped weights remain dropped for the entirety of the forward and backward pass. The result is similar to variational dropout, which applies the same dropout mask to recurrent connections within the LSTM by performing dropout on $h_{t-1}$, except that the dropout is applied to the recurrent weights. DropConnect could also be used on the non-recurrent weights of the LSTM $[W^i, W^f, W^o]$ though our focus was on preventing overfitting on the recurrent connection.

## 3 OPTIMIZATION

SGD is among the most popular methods for training deep learning models across various modalities including computer vision, natural language processing, and reinforcement learning. The training of deep networks can be posed as a non-convex empirical risk minimization problem

$$\min_{w} \quad \frac{1}{N} \sum_{i=1}^{N} f_i(w),$$

where $f_i$ is the loss function for the $i^{th}$ data point, $w$ are the weights of the network, and the expectation is taken over the data. In this context, given a sequence of learning rates, $\gamma_k$, SGD iteratively takes steps of the form

$$w_{k+1} = w_k - \gamma_k \hat{\nabla} f(w_k), \qquad (1)$$

where the subscript denotes the iteration number and the $\hat{\nabla}$ denotes a stochastic gradient that may be computed on a minibatch of data points. SGD demonstrably performs well in practice and also possesses several attractive theoretical properties such as linear convergence (Bottou et al., 2016), saddle point avoidance (Panageas & Piliouras, 2016) and better generalization performance (Hardt et al., 2015). For the specific task of neural language modeling, traditionally SGD without momentum has been found to outperform other algorithms such as momentum SGD (Sutskever et al., 2013), Adam (Kingma & Ba, 2014), Adagrad (Duchi et al., 2011) and RMSProp (Tieleman & Hinton, 2012) by a statistically significant margin.

Motivated by this observation, we investigate averaged SGD (AvSGD) to further improve the training process. AvSGD has been analyzed in depth theoretically and many surprising results have been shown including its asymptotic second-order convergence (Polyak & Juditsky, 1992; Mandt et al., 2017). AvSGD takes steps identical to equation (1) but instead of returning the last iterate as the solution, returns $\frac{1}{(K-T+1)} \sum_{i=T}^{K} w_i$, where $K$ is the total number of iterations and $T < K$ is a user-specified averaging trigger.

Despite its theoretical appeal, AvSGD has found limited practical use in training of deep networks. This may be in part due to unclear tuning guidelines for the learning-rate schedule $\gamma_k$ and averaging trigger $T$. If the averaging is triggered too soon, the efficacy of the method is impacted, and if it is triggered too late, many additional iterations may be needed to converge to the solution. In this section, we describe a non-monotonically triggered variant of AvSGD (NT-AvSGD), which obviates the need for tuning $T$. Further, the algorithm uses a constant learning rate throughout the experiment and hence no further tuning is necessary for the decay scheduling.

Ideally, averaging needs to be triggered when the SGD iterates converge to a steady-state distribution (Mandt et al., 2017). This is roughly equivalent to the convergence of SGD to a neighborhood around a solution. In the case of SGD, certain learning-rate reduction strategies such as the step-wise strategy analogously reduce the learning rate by a fixed quantity at such a point. A common strategy employed in language modeling is to reduce the learning rates by a fixed proportion when the performance of the model's primary metric (such as perplexity) worsens or stagnates. Along the same lines, one could make a triggering decision based on the performance of the model on the

---

**Algorithm 1** Non-monotonically Triggered AvSGD (NT-AvSGD)

---

**Inputs:** Initial point $w_0$, learning rate $\gamma$, logging interval $L$, non-monotone interval $n$.

1: Initialize $k \leftarrow 0$, $t \leftarrow 0$, $T \leftarrow 0$, `logs` $\leftarrow$ `[]`
2: **while** stopping criterion not met **do**
3:     Compute stochastic gradient $\hat{\nabla} f(w_k)$ and take SGD step (1).
4:     **if** $\text{mod}(k, L) = 0$ and $T = 0$ **then**
5:         Compute validation perplexity $v$.
6:         **if** $t > n$ and $v > \min\limits_{l \in \{0, \cdots, t-n-1\}} \texttt{logs[l]}$ **then**
7:             Set $T \leftarrow k$
8:         **end if**
9:         Append $v$ to `logs`
10:       $t \leftarrow t + 1$
11:     **end if**
12:     $k \leftarrow k + 1$
13: **end while**
**return** $\frac{\sum_{i=T}^{k} w_i}{(k-T+1)}$

---

validation set. However, instead of averaging immediately after the validation metric worsens, we propose a non-monotonic criterion that conservatively triggers the averaging when the validation metric fails to improve for multiple cycles; see Algorithm 1. Given that the choice of triggering is irreversible, this conservatism ensures that the randomness of training does not play a major role in the decision. Analogous strategies have also been proposed for learning-rate reduction in SGD (Keskar & Saon, 2015).

While the algorithm introduces two additional hyperparameters, the logging interval $L$ and non-monotone interval $n$, we found that setting $L$ to be the number of iterations in an epoch and $n = 5$ worked well across various models and data sets. As such, we use this setting in all of our NT-AvSGD experiments in the following section and demonstrate that it achieves better training outcomes as compared to SGD.

## 4   Extended regularization techniques

In addition to the regularization and optimization techniques above, we explored additional regularization techniques that aimed to improve data efficiency during training and to prevent overfitting of the RNN model.

### 4.1   Variable length backpropagation sequences

Given a fixed sequence length that is used to break a data set into fixed length batches, the data set is not efficiently used. To illustrate this, imagine being given 100 elements to perform backpropagation through with a fixed backpropagation through time (BPTT) window of 10. Any element divisible by 10 will never have any elements to backprop into, no matter how many times you may traverse the data set. Indeed, the backpropagation window that each element receives is equal to $i \bmod 10$ where $i$ is the element's index. This is data inefficient, preventing $\frac{1}{10}$ of the data set from ever being able to improve itself in a recurrent fashion, and resulting in $\frac{8}{10}$ of the remaining elements receiving only a partial backpropagation window compared to the full possible backpropagation window of length 10.

To prevent such inefficient data usage, we randomly select the sequence length for the forward and backward pass in two steps. First, we select the base sequence length to be seq with probability $p$ and $\frac{\text{seq}}{2}$ with probability $1 - p$, where $p$ is a high value approaching 1. This spreads the starting point for the BPTT window beyond the base sequence length. We then select the sequence length according to $\mathcal{N}(\text{seq}, s)$, where seq is the base sequence length and $s$ is the standard deviation. This jitters the starting point such that it doesn't always fall on a specific word divisible by seq or $\frac{\text{seq}}{2}$. From these, the sequence length more efficiently uses the data set, ensuring that when given enough

epochs all the elements in the data set experience a full BPTT window, while ensuring the average sequence length remains around the base sequence length for computational efficiency.

During training, we rescale the learning rate depending on the length of the resulting sequence compared to the original specified sequence length. The rescaling step is necessary as sampling arbitrary sequence lengths with a fixed learning rate favors short sequences over longer ones. This linear scaling rule has been noted as important for training large scale minibatch SGD without loss of accuracy (Goyal et al., 2017) and is a component of unbiased truncated backpropagation through time (Tallec & Ollivier, 2017).

## 4.2 VARIATIONAL DROPOUT

In standard dropout, a new binary dropout mask is sampled each and every time the dropout function is called. New dropout masks are sampled even if the given connection is repeated, such as the input $x_0$ to an LSTM at timestep $t = 0$ receiving a different dropout mask than the input $x_1$ fed to the same LSTM at $t = 1$. A variant of this, variational dropout (Gal & Ghahramani, 2016), samples a binary dropout mask only once upon the first call and then to repeatedly use that locked dropout mask for all repeated connections within the forward and backward pass.

While we propose using DropConnect rather than variational dropout to regularize the hidden-to-hidden transition within an RNN, we use variational dropout for all other dropout operations, specifically using the same dropout mask for all inputs and outputs of the LSTM within a given forward and backward pass. Each example within the minibatch uses a unique dropout mask, rather than a single dropout mask being used over all examples, ensuring diversity in the elements dropped out.

## 4.3 EMBEDDING DROPOUT

Following Gal & Ghahramani (2016), we employ embedding dropout. This is equivalent to performing dropout on the embedding matrix at a word level, where the dropout is broadcast across all the word vector's embedding. The remaining non-dropped-out word embeddings are scaled by $\frac{1}{1-p_e}$ where $p_e$ is the probability of embedding dropout. As the dropout occurs on the embedding matrix that is used for a full forward and backward pass, this means that all occurrences of a specific word will disappear within that pass, equivalent to performing variational dropout on the connection between the one-hot embedding and the embedding lookup.

## 4.4 WEIGHT TYING

Weight tying (Inan et al., 2016; Press & Wolf, 2016) shares the weights between the embedding and softmax layer, substantially reducing the total parameter count in the model. The technique has theoretical motivation (Inan et al., 2016) and prevents the model from having to learn a one-to-one correspondence between the input and output, resulting in substantial improvements to the standard LSTM language model.

## 4.5 INDEPENDENT EMBEDDING SIZE AND HIDDEN SIZE

In most natural language processing tasks, both pre-trained and trained word vectors are of relatively low dimensionality—frequently between 100 and 400 dimensions in size. Most previous LSTM language models tie the dimensionality of the word vectors to the dimensionality of the LSTM's hidden state. Even if reducing the word embedding size was not beneficial in preventing overfitting, the easiest reduction in total parameters for a language model is reducing the word vector size. To achieve this, the first and last LSTM layers are modified such that their input and output dimensionality respectively are equal to the reduced embedding size.

## 4.6 ACTIVATION REGULARIZATION (AR) AND TEMPORAL ACTIVATION REGULARIZATION (TAR)

$L_2$-regularization is often used on the weights of the network to control the norm of the resulting model and reduce overfitting. In addition, $L_2$ decay can be used on the individual unit activations and on the difference in outputs of an RNN at different time steps; these strategies labeled as activation

regularization (AR) and temporal activation regularization (TAR) respectively (Merity et al., 2017). AR penalizes activations that are significantly larger than 0 as a means of regularizing the network. Concretely, AR is defined as

$$\alpha \, L_2(m \odot h_t)$$

where $m$ is the dropout mask, $L_2(\cdot) = \|\cdot\|_2$, $h_t$ is the output of the RNN at timestep $t$, and $\alpha$ is a scaling coefficient. TAR falls under the broad category of *slowness* regularizers (Hinton, 1989; Földiák, 1991; Luciw & Schmidhuber, 2012; Jonschkowski & Brock, 2015) which penalize the model from producing large changes in the hidden state. Using the notation from AR, TAR is defined as

$$\beta \, L_2(h_t - h_{t+1})$$

where $\beta$ is a scaling coefficient. As in Merity et al. (2017), the AR and TAR loss are only applied to the output of the final RNN layer as opposed to being applied to all layers.

## 5 EXPERIMENT DETAILS

For evaluating the impact of these approaches, we perform language modeling over a preprocessed version of the Penn Treebank (PTB) (Mikolov et al., 2010) and the WikiText-2 (WT2) data set (Merity et al., 2016).

**PTB:** The Penn Treebank data set has long been a central data set for experimenting with language modeling. The data set is heavily preprocessed and does not contain capital letters, numbers, or punctuation. The vocabulary is also capped at 10,000 unique words, quite small in comparison to most modern datasets, which results in a large number of out of vocabulary (OoV) tokens.

**WT2:** WikiText-2 is sourced from curated Wikipedia articles and is approximately twice the size of the PTB data set. The text is tokenized and processed using the Moses tokenizer (Koehn et al., 2007), frequently used for machine translation, and features a vocabulary of over 30,000 words. Capitalization, punctuation, and numbers are retained in this data set.

All experiments use a three-layer LSTM model with 1150 units in the hidden layer and an embedding of size 400. The loss was averaged over all examples and timesteps. All embedding weights were uniformly initialized in the interval $[-0.1, 0.1]$ and all other weights were initialized between $[-\frac{1}{\sqrt{H}}, \frac{1}{\sqrt{H}}]$, where $H$ is the hidden size.

For training the models, we use the NT-AvSGD algorithm discussed in the previous section for 750 epochs with $L$ equivalent to one epoch and $n = 5$. We use a batch size of 80 for WT2 and 40 for PTB. Empirically, we found relatively large batch sizes (e.g., 40-80) performed better than smaller sizes (e.g., 10-20) for NT-AvSGD. After completion, we run AvSGD with $T = 0$ and hot-started $w_0$ as a fine-tuning step to further improve the solution. For this fine-tuning step, we terminate the run using the same non-monotonic criterion detailed in Algorithm 1.

We carry out gradient clipping with maximum norm 0.25 and use an initial learning rate of 30 for all experiments. We use a random BPTT length which is $\mathcal{N}(70, 5)$ with probability 0.95 and $\mathcal{N}(35, 5)$ with probability 0.05. The values used for dropout on the word vectors, the output between LSTM layers, the output of the final LSTM layer, and embedding dropout where $(0.4, 0.3, 0.4, 0.1)$ respectively. For the weight-dropped LSTM, a dropout of 0.5 was applied to the recurrent weight matrices. For WT2, we increase the input dropout to 0.65 to account for the increased vocabulary size. For all experiments, we use AR and TAR values of 2 and 1 respectively, and tie the embedding and softmax weights. These hyperparameters were chosen through trial and error and we expect further improvements may be possible if a fine-grained hyperparameter search were to be conducted. In the results, we abbreviate our approach as AWD-LSTM for AvSGD Weight-Dropped LSTM. The code for reproducing our results is open sourced and available at `https://github.com/salesforce/awd-lstm-lm`.

| Model | Parameters | Validation | Test |
|---|---|---|---|
| Mikolov & Zweig (2012) - KN-5 | 2M[‡] | – | 141.2 |
| Mikolov & Zweig (2012) - KN5 + cache | 2M[‡] | – | 125.7 |
| Mikolov & Zweig (2012) - RNN | 6M[‡] | – | 124.7 |
| Mikolov & Zweig (2012) - RNN-LDA | 7M[‡] | – | 113.7 |
| Mikolov & Zweig (2012) - RNN-LDA + KN-5 + cache | 9M[‡] | – | 92.0 |
| Zaremba et al. (2014) - LSTM (medium) | 20M | 86.2 | 82.7 |
| Zaremba et al. (2014) - LSTM (large) | 66M | 82.2 | 78.4 |
| Gal & Ghahramani (2016) - Variational LSTM | 20M | – | 78.6 |
| Gal & Ghahramani (2016) - Variational LSTM | 66M | – | 73.4 |
| Kim et al. (2016) - CharCNN | 19M | – | 78.9 |
| Merity et al. (2016) - Pointer Sentinel-LSTM | 21M | 72.4 | 70.9 |
| Grave et al. (2016) - LSTM | – | – | 82.3 |
| Grave et al. (2016) - LSTM + continuous cache pointer | – | – | 72.1 |
| Inan et al. (2016) - Variational LSTM (tied) + augmented loss | 24M | 75.7 | 73.2 |
| Inan et al. (2016) - Variational LSTM (tied) + augmented loss | 51M | 71.1 | 68.5 |
| Zilly et al. (2016) - Variational RHN (tied) | 23M | 67.9 | 65.4 |
| Zoph & Le (2016) - NAS Cell (tied) | 25M | – | 64.0 |
| Zoph & Le (2016) - NAS Cell (tied) | 54M | – | 62.4 |
| Melis et al. (2017) - 4-layer skip connection LSTM (tied) | 24M | 60.9 | 58.3 |
| AWD-LSTM - 3-layer LSTM (tied) | 24M | 60.0 | 57.3 |
| AWD-LSTM - 3-layer LSTM (tied) + continuous cache pointer | 24M | 53.9 | 52.8 |

Table 1: Single model perplexity on validation and test sets for the Penn Treebank language modeling task. Parameter numbers with ‡ are estimates based upon our understanding of the model and with reference to (Merity et al., 2016). Models noting *tied* use weight tying on the embedding and softmax weights. Our model, AWD-LSTM, stands for AvSGD Weight-Dropped LSTM.

| Model | Parameters | Validation | Test |
|---|---|---|---|
| Inan et al. (2016) - Variational LSTM (tied) | 28M | 92.3 | 87.7 |
| Inan et al. (2016) - Variational LSTM (tied) + augmented loss | 28M | 91.5 | 87.0 |
| Grave et al. (2016) - LSTM | – | – | 99.3 |
| Grave et al. (2016) - LSTM + continuous cache pointer | – | – | 68.9 |
| Melis et al. (2017) - 1-layer LSTM (tied) | 24M | 69.3 | 65.9 |
| Melis et al. (2017) - 2-layer skip connection LSTM (tied) | 24M | 69.1 | 65.9 |
| AWD-LSTM - 3-layer LSTM (tied) | 33M | 68.6 | 65.8 |
| AWD-LSTM - 3-layer LSTM (tied) + continuous cache pointer | 33M | 53.8 | 52.0 |

Table 2: Single model perplexity over WikiText-2. Models noting *tied* use weight tying on the embedding and softmax weights. Our model, AWD-LSTM, stands for AvSGD Weight-Dropped LSTM.

## 6 EXPERIMENTAL ANALYSIS

We present the single-model perplexity results for both our models (AWD-LSTM) and other competitive models in Table 1 and 2 for PTB and WT2 respectively [1]. On both data sets we improve the state-of-the-art, with our vanilla LSTM model beating the state of the art by approximately 1 unit on PTB and 0.1 units on WT2.

In comparison to other recent state-of-the-art models, our model uses a vanilla LSTM. Zilly et al. (2016) propose the recurrent highway network, which extends the LSTM to allow multiple hidden state updates per timestep. Zoph & Le (2016) use a reinforcement learning agent to generate an RNN cell tailored to the specific task of language modeling, with the cell far more complex than the LSTM.

---

[1] During the course of code refactoring, we discovered better hyperparameter settings for the dropout values that led to an additional ∼ 0.5 perplexity improvement on both PTB and WikiText-2; see our code release for specifics. The values reported here and in the analysis section pertain to the original configuration.

Independently of our work, Melis et al. (2017) apply extensive hyperparameter search to an LSTM based language modeling implementation, analyzing the sensitivity of RNN based language models to hyperparameters. Unlike our work, they use a modified LSTM, which caps the input gate $i_t$ to be $\min(1 - f_t, i_t)$, use Adam with $\beta_1 = 0$ rather than SGD or AvSGD, use skip connections between LSTM layers, and use a black box hyperparameter tuner for exploring models and settings. Of particular interest is that their hyperparameters were tuned individually for each data set compared to our work which shared almost all hyperparameters between PTB and WT2, including the embedding and hidden size for both data sets. Due to this, they used less model parameters than our model and found shallow LSTMs of one or two layers worked best for WT2.

Like our work, Melis et al. (2017) find that the underlying LSTM architecture can be highly effective compared to complex custom architectures when well tuned hyperparameters are used. The approaches used in our work and (Melis et al., 2017) may be complementary and would be worth exploration.

## 6.1 POINTER MODELS

In past work, pointer based attention models have been shown to be highly effective in improving language modeling (Merity et al., 2016; Grave et al., 2016). Given such substantial improvements to the underlying neural language model, it remained an open question as to how effective pointer augmentation may be, especially when improvements such as weight tying may act in mutually exclusive ways.

The neural cache model (Grave et al., 2016) can be added on top of a pre-trained language model at negligible cost. The neural cache stores the previous hidden states in memory cells and then uses a simple convex combination of the probability distributions suggested by the cache and the language model for prediction. The cache model has three hyperparameters: the memory size (window) for the cache, the coefficient of the combination (which determines how the two distributions are mixed), and the flatness of the cache distribution. All of these are tuned on the validation set once a trained language model has been obtained and require no training by themselves, making it quite inexpensive to use. The tuned values for these hyperparameters were $(2000, 0.1, 1.0)$ for PTB and $(3785, 0.1279, 0.662)$ for WT2 respectively.

In Tables 1 and 2, we show that the model further improves the perplexity of the language model by as much as 6 perplexity points for PTB and 11 points for WT2. While this is smaller than the gains reported in Grave et al. (2016), which used an LSTM without weight tying, this is still a substantial drop. Given the simplicity of the neural cache model, and the lack of any trained components, these results suggest that existing neural language models remain fundamentally lacking, failing to capture long term dependencies or remember recently seen words effectively.

To understand the impact the pointer had on the model, specifically the validation set perplexity, we detail the contribution that each word has on the cache model's overall perplexity in Table 3. We compute the sum of the total difference in the loss function value (i.e., log perplexity) between the LSTM-only and LSTM-with-cache models for the target words in the validation portion of the WikiText-2 data set. We present results for the sum of the difference as opposed to the mean since the latter undesirably overemphasizes infrequently occurring words for which the cache helps significantly and ignores frequently occurring words for which the cache provides modest improvements that cumulatively make a strong contribution.

The largest cumulative gain is in improving the handling of <unk> tokens, though this is over 11540 instances. The second best improvement, approximately one fifth the gain given by the <unk> tokens, is for Meridian, yet this word only occurs 161 times. This indicates the cache still helps significantly even for relatively rare words, further demonstrated by Churchill, Blythe, or Sonic. The cache is not beneficial when handling frequent word categories, such as punctuation or stop words, for which the language model is likely well suited. These observations motivate the design of a cache framework that is more aware of the relative strengths of the two models.

## 6.2 AWD-QRNN

Several architectures for learning sequential data based on convolutions, instead of recurrences, have been proposed recently. We briefly mention experiments on the same language modeling us-

| Word | Count | $\Delta$loss | Word | Count | $\Delta$loss |
|------|-------|--------------|------|-------|--------------|
| . | 7632 | -696.45 | <unk> | 11540 | 5047.34 |
| , | 9857 | -687.49 | Meridian | 161 | 1057.78 |
| of | 5816 | -365.21 | Churchill | 137 | 849.43 |
| = | 2884 | -342.01 | - | 67 | 682.15 |
| to | 4048 | -283.10 | Blythe | 97 | 554.95 |
| in | 4178 | -222.94 | Sonic | 75 | 543.85 |
| <eos> | 3690 | -216.42 | Richmond | 101 | 429.18 |
| and | 5251 | -215.38 | Starr | 74 | 416.52 |
| the | 12481 | -209.97 | Australian | 234 | 366.36 |
| a | 3381 | -149.78 | Pagan | 54 | 365.19 |
| " | 2540 | -127.99 | Asahi | 39 | 316.24 |
| that | 1365 | -118.09 | Japanese | 181 | 295.97 |
| by | 1252 | -113.05 | Hu | 43 | 285.58 |
| was | 2279 | -107.95 | Hedgehog | 29 | 266.48 |
| ) | 1101 | -94.74 | Burma | 35 | 263.65 |
| with | 1176 | -93.01 | 29 | 92 | 260.88 |
| for | 1215 | -87.68 | Mississippi | 72 | 241.59 |
| on | 1485 | -81.55 | German | 108 | 241.23 |
| as | 1338 | -77.05 | mill | 67 | 237.76 |
| at | 879 | -59.86 | Cooke | 33 | 231.11 |

Table 3: The sum total difference in loss (log perplexity) that a given word results in over all instances in the validation data set of WikiText-2 when the continuous cache pointer is introduced. The right column contains the words with the twenty best improvements (i.e., where the cache was advantageous), and the left column the twenty most deteriorated (i.e., where the cache was disadvantageous).

| Model | PTB | | WT2 | |
|-------|-----------|------|-----------|------|
| | Validation | Test | Validation | Test |
| AWD-LSTM only training | 60.7 | 58.3 | 69.1 | 66.0 |
| + fine tune | 60.0 | 57.3 | 68.6 | 65.8 |
| + fine tune + continuous cache pointer | 53.9 | 52.8 | 53.8 | 52.0 |
| QRNN-LSTM only training | 60.6 | 58.3 | 69.3 | 66.8 |
| + fine tune | 59.1 | 56.7 | 68.5 | 65.9 |
| + fine tune + continuous cache pointer | 53.4 | 52.6 | 53.6 | 52.1 |

Table 4: Comparison of AWD-LSTM and AWD-QRNN for the same model size on the PTB and WikiText-2 data sets.

ing quasi-recurrent neural networks (QRNNs) (Bradbury et al., 2016) instead of LSTMs; we label this setup the AWD-QRNN. As in the case of AWD-LSTM, we regularize the network through weight, embedding and variational dropouts along with variable sequence lengths, weight tying, AR and TAR. The networks were designed such that they had the same number of parameters as their LSTM counterparts and were trained using NT-AvSGD. Despite the same size of the network, QRNNs were $2 - 4\times$ faster per epoch as compared to their LSTM counterparts and required fewer epochs to converge. We report the results in Table 4. As is evident from the table, the QRNN model achieves comparable results to the LSTM suggesting the generality of the proposed regularization techniques. Interestingly, the hyperparameter values for the various regularization components, including the optimization procedure, needed minimal changes from the LSTM to the QRNN models for competitive performance. For full details and hyperparameters, refer to the released code.

## 6.3 MODEL ABLATION ANALYSIS

In Table 5, we present the values of validation and testing perplexity for different variants of our best-performing LSTM model. Each variant removes a form of optimization or regularization.

| Model | PTB | | WT2 | |
|---|---|---|---|---|
| | **Validation** | **Test** | **Validation** | **Test** |
| AWD-LSTM (tied) | 60.0 | 57.3 | 68.6 | 65.8 |
| – fine-tuning | 60.7 | 58.8 | 69.1 | 66.0 |
| – NT-AvSGD | 66.3 | 63.7 | 73.3 | 69.7 |
| – variable sequence lengths | 61.3 | 58.9 | 69.3 | 66.2 |
| – embedding dropout | 65.1 | 62.7 | 71.1 | 68.1 |
| – weight decay | 63.7 | 61.0 | 71.9 | 68.7 |
| – AR/TAR | 62.7 | 60.3 | 73.2 | 70.1 |
| – full sized embedding | 68.0 | 65.6 | 73.7 | 70.7 |
| – weight-dropping | 71.1 | 68.9 | 78.4 | 74.9 |

Table 5: Model ablations for our best LSTM models reporting results over the validation and test set on Penn Treebank and WikiText-2. Ablations are split into optimization and regularization variants, sorted according to the achieved validation perplexity on WikiText-2.

The first two variants deal with the optimization of the language models while the rest deal with the regularization. For the model using SGD with learning rate reduced by 2 using the same nonmonotonic fashion, there is a significant degradation in performance. This stands as empirical evidence regarding the benefit of averaging of the iterates. Using a monotonic criterion instead also hampered performance. Similarly, the removal of the fine-tuning step expectedly also degrades the performance. This step helps improve the estimate of the minimizer by resetting the memory of the previous experiment. While this process of fine-tuning can be repeated multiple times, we found little benefit in repeating it more than once.

The removal of regularization strategies paints a similar picture; the inclusion of all of the proposed strategies was pivotal in ensuring state-of-the-art performance. The most extreme perplexity jump was in removing the hidden-to-hidden LSTM regularization provided by the weight-dropped LSTM. Without such hidden-to-hidden regularization, perplexity rises substantially, up to 11 points. This is in line with previous work showing the necessity of recurrent regularization in state-of-the-art models (Gal & Ghahramani, 2016; Inan et al., 2016).

We also experiment with static sequence lengths which we had hypothesized would lead to inefficient data usage. This also worsens the performance by approximately one perplexity unit. Next, we experiment with reverting to matching the sizes of the embedding vectors and the hidden states. This significantly increases the number of parameters in the network (to 43M in the case of PTB and 70M for WT2) and leads to degradation by almost 8 perplexity points, which we attribute to overfitting in the word embeddings. While this could potentially be improved with more aggressive regularization, the computational overhead involved with substantially larger embeddings likely outweighs any advantages. Finally, we experiment with the removal of embedding dropout, AR/TAR and weight decay. In all of the cases, the model suffers a perplexity increase of 2–6 points which we hypothesize is due to insufficient regularization in the network.

## 7 CONCLUSION

In this work, we discuss regularization and optimization strategies for neural language models. We propose the weight-dropped LSTM, a strategy that uses a DropConnect mask on the hidden-to-hidden weight matrices, as a means to prevent overfitting across the recurrent connections. Further, we investigate the use of averaged SGD with a non-monontonic trigger for training language models and show that it outperforms SGD by a significant margin. We investigate other regularization strategies including the use of variable BPTT length and achieve a new state-of-the-art perplexity on the PTB and WikiText-2 data sets. Our models outperform custom-built RNN cells and complex regularization strategies that preclude the possibility of using optimized libraries such as the NVIDIA cuDNN LSTM. We explore the use of a neural cache in conjunction with our proposed model and show that this further improves the performance, thus attaining an even lower state-of-the-art perplexity. We also explore the viability of using the proposed regularization and

optimization strategies in the context of a quasi-recurrent neural network (QRNN) and demonstrate comparable performance to the LSTM counterpart. While the regularization and optimization strategies proposed are demonstrated on the task of language modeling, we anticipate that they would be generally applicable across other sequence learning tasks.

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
