# OpenReview forum: "Regularizing and Optimizing LSTM Language Models"
_ICLR.cc/2018/Conference — Accept (Poster)_

### Official Review · AnonReviewer3 · 2017-11-24
**State of the art on small word level language modelling datasets**

**Rating:** 7
**Confidence:** 4

**Review:**

The paper sets a new state of the art on word level language modelling on the Penn Treebank and Wikitext-2 datasets using various optimization and regularization techniques. These already very good results are further improved, by a large margin, using a Neural Cache.

The paper is well written, easy to follow and the results speak for themselves. One possible criticism is that the experimental methodology does not allow for reliable conclusions to be drawn about contributions of all different techniques, because they seem to have been evaluated at a single hyperparameter setting (that was hand tuned for the full model?).

A variant on the Averaged SGD method is proposed. This so called NT-ASGD optimizer switches to averaging mode based on recent validation losses. I would have liked to see a more thorough assessment of NT-ASGD, especially against well tuned SGD.

I particularly liked Figure 3 which shows how the Neural Cache makes the model much better at handling rare words and UNK (!) at the expense of very common words. Speaking of the Neural Cache, a natural baseline would have been dynamic evaluation.

All in all, the paper is a solid contribution which deserves to be accepted. It could become even better, were the experiments to tease the various factors apart.

---

> ### Author Response · Authors · 2017-12-22
> **Thank you for your review.**
>
> “Speaking of the Neural Cache, a natural baseline would have been dynamic evaluation.”
> We agree. Incidentally, there is a recent manuscript which applies dynamic evaluation to the AWD-LSTM framework resulting in lower perplexity values. We will refrain from providing additional details about this work to protect the double blind anonymity.
>
>
> “This so called NT-ASGD optimizer switches to averaging mode based on recent validation losses. I would have liked to see a more thorough assessment of NT-ASGD, especially against well tuned SGD.”
>
> We conducted a few experiments comparing NT-ASGD and SGD for different initial learning rates and tabulate the results below. In the SGD experiments, we use the same non-monotonically triggered criterion but instead of averaging the iterates, use it to reduce the learning rate by 4 (we also tried 2, 5 and 10).
>
> +-----------+------------------+
> |      LR    |  SGD  | ASGD|
> +-----------+-------+----------+
> |       30   | 61.52 | 58.47 |
> +-----------+-------+----------+
> |       40   | 61.77 | 59.98 |
> +-----------+-------+----------+
> |       50   | 63.30 | 64.32 |
> +-----------+-------+----------+
> |       60   | 64.86 | 69.78 |
> +-----------+-------+----------+

---

### Official Review · AnonReviewer2 · 2017-11-27
**Evaluation of a number of heuristics to improve LSTM language models.**

**Rating:** 7
**Confidence:** 4

**Review:**

Clearly presented paper, including a number of reasonable techniques to improve LSTM-LMs. The proposed techniques are heuristic, but are reasonable and appear to yield improvements in perplexity. Some specific comments follow.

re. "ASGD" for Averaged SGD: ASGD usually stands for Asynchronous SGD, have the authors considered an alternative acronym? AvSGD?

re. Optimization criterion on page 2, note that SGD is usually taken to minimizing expected loss, not just empirical loss (Bottou thesis 1991).

Is there any theoretical analysis of convergence for Averaged SGD?

re. paragraph starting with "To prevent such inefficient data usage, we randomly select the sequence length for the forward and backward pass in two steps": the explanation is a bit unclear. What is the "base sequence length" exactly? Also, re. the motivation above this paragraph, I'm not sure what "elements" really refers to, though I can guess.

What is the number of training tokens of the datasets used, PTB and WT2?

Can the authors provide more explanation for what "neural cache models" are, and how they relate to "pointer models"?

Why do the sections "Pointer models", "Ablation analysis", and "AWD-QRNN" come after the Experiments section?

---

> ### Author Response · Authors · 2017-12-22
> **Thank you for your review.**
>
> "re. "ASGD" for Averaged SGD: ASGD usually stands for Asynchronous SGD, have the authors considered an alternative acronym? AvSGD?"
>
> Agreed! We have made the suggested change.
>
> "re. Optimization criterion on page 2, note that SGD is usually taken to minimizing expected loss, not just empirical loss (Bottou thesis 1991)."
>
> We agree and have made the suggested change.
>
> "Is there any theoretical analysis of convergence for Averaged SGD?"
>
> Averaged SGD has been studied quite extensively; other than standard guarantees of convergence, two theoretical contributions stand out. First, in “Acceleration of stochastic approximation by averaging” Polyak and Juditsky show that, under circumstances, averaged SGD achieves the same asymptotic convergence rate as a second-order stochastic method. Second, in “Stochastic Gradient Descent as Approximate Bayesian Inference”, Mandt et. al. show that averaged SGD reduces the variance of the (noisy) SGD iterates around the minimizer of the loss function.
>
> "re. paragraph starting with "To prevent such inefficient data usage, we randomly select the sequence length for the forward and backward pass in two steps": the explanation is a bit unclear. What is the "base sequence length" exactly? Also, re. the motivation above this paragraph, I'm not sure what "elements" really refers to, though I can guess."
>
> We wanted to keep it generic so used elements rather than words but that may not have been best choice. The main aim is to prevent the model from seeing the data in the exact same batches each time. This is not a problem in many other tasks due to shuffling - but shuffling can’t be done when the data is sequential. The base sequence length for both PTB and WT-2 are 70 tokens though that is a hyperparameter that can be freely modified.
>
> "What is the number of training tokens of the datasets used, PTB and WT2?"
>
> PTB has 887k training tokens, WT2 has 2088k training tokens.
>
> "Can the authors provide more explanation for what "neural cache models" are, and how they relate to "pointer models"?"
>
> They are quite similar; in our setup, the cache models are used atop an existing trained language model. The model uses hidden states from the previous tokens to point and in conjunction with the softmax, determines the next token. On the other hand, pointer models are trained in conjunction with the language model. Both point to previous tokens as a way to determine probability distributions for the next word.
>
> "Why do the sections "Pointer models", "Ablation analysis", and "AWD-QRNN" come after the Experiments section?"
>
> We agree that they seem out of place and have reordered the sections.

---

### Official Review · AnonReviewer1 · 2017-12-05
**This paper proposes regularization strategies for word LSTM-based language models achieving state-of-the-art results on the PennTree Bank and the Wiki-Text2 LM tasks. A variant of  Average-SGD (ASGD) where the threshold for determining which iterates to average is also proposed, yielding better results than SGD.**

**Rating:** 7
**Confidence:** 5

**Review:**

This is a well-written paper that proposes regularization and optimization strategies for word-based language modeling tasks.   The authors propose the use of DropConnect  on the hidden-hidden connections as a regularization method, in order to take advantage of high-speed LSTM implementations via the cuDNN LSTM libraries from NVIDIA.  The  focus of this work is on the prevention of overfitting on the recurrent connections of the LSTM.  The authors explore a variant of Average-SGD (NT-ASGD) as an optimization strategy which eliminates the need for tuning the average trigger and uses a constant learning rate.  Averaging is triggered when the validation loss worsens or stagnates for a few cycles, leading to two new hyper parameters: logging interval and non-monotone interval.  Other forms of well-know regularization methods were applied to the non-recurrent connections, input, output and embedding matrices.

As the authors point out, all the methods used in this paper have been proposed before and theoretical convergence explained. The novelty of this work lies in its successful application to the language modeling task achieving state-of-the-art results.

On the PTB task, the proposed AWD-LSTM achieves a perplexity of 57.3 vs 58.3 (Melis et al 2017) and almost the same perplexity as Melis et el. on the Wiki-Text2 task (65.8 vs 65.9).  The addition of a cache model provides significant gains on both tasks.

It would be useful, if authors had explored the behavior of the  AWD-LSTM algorithm with respect to various hyper parameters  and provided a few insights towards their choices for other large vocabulary language modeling tasks (1 million vocabulary sizes).

Similarly, the choice of the average trigger and number of cycles seem arbitrary -  it would have been good to see a graph over a range of values, showing their impact on the model's performance.

A 3-layer LSTM has been used for the experiments  - how was this choice made?  What is the impact of this algorithm if the net was a 2-layer net as is typical in most large-scale LMs?

Table 3 is interesting to see how the cache model helps with rare words  and as such has applications in key word spotting tasks. Were the hyper parameters of the cache tuned to perform better on rare words?  More details on the design of the cache model would have been useful.

You state that the gains obtained using the cache model were far less than what was obtained in Graves et al 2016 - what do you attribute this to?

Ablation analysis in Table 4 is very useful - in particular it shows how lack of regularization of the recurrent connections can lead to maximum degradation in performance.

Most of the results in this paper have been based on one choice of various model parameters. Given the emperical nature of this work, it would have made the paper even clearer if an analysis of their choices were presented.  Overall, it would be beneficial to the MLP community to see this paper accepted in the conference.

---

> ### Author Response · Authors · 2017-12-22
> **Thank you for your review.**
>
> "It would be useful, if authors had explored the behavior of the  AWD-LSTM algorithm with respect to various hyper parameters  and provided a few insights towards their choices for other large vocabulary language modeling tasks (1 million vocabulary sizes).  "
>
> We have done preliminary experiments with data sets with large vocabulary sizes (such as WikiText-103 and the Google One Billion Word Corpus). Due to the large softmax costs associated with an increased vocabulary, an adaptive or hierarchical softmax is indispensable. In this case, tying the word vectors and softmax weights is non-trivial. Using a naive tying approach and the AWD-QRNN architecture described in the paper, we were able to train WikiText-103 to state-of-the-art performance and have received favorable initial results for One Billion Word corpus as well. This line of research warrants more work related to scalability and convergence and as such we will be continuing our investigation and analysis.
>
> "Similarly, the choice of the average trigger and number of cycles seem arbitrary -  it would have been good to see a graph over a range of values, showing their impact on the model's performance."
>
> We have carried out a sensitivity experiment and tabulate the results below. In particular, we vary the number of cycles from 2 to 10 and report the testing perplexity for AWD-QRNN on the PTB data set along with the epoch at which the averaging was triggered. The final perplexity is fairly insensitive to precise specification of the cycle length; this observation is true on the other models as well.
>
> +-----------------+--------------------+-----------------+
> |Interval Len.|   T (epochs)    | Test Perp.   |
> +-----------------+--------------------+-----------------+
> |          2          |           58           |      58.74      |
> +-----------------+--------------------+-----------------+
> |          3          |           58           |      58.74      |
> +-----------------+--------------------+-----------------+
> |          4          |           68           |      58.47      |
> +-----------------+--------------------+-----------------+
> |          5          |           68           |      58.47      |
> +-----------------+--------------------+-----------------+
> |          6          |           68           |      58.47      |
> +-----------------+--------------------+-----------------+
> |          7          |           71           |      58.42      |
> +-----------------+--------------------+-----------------+
> |          8          |           72           |      58.37      |
> +-----------------+--------------------+-----------------+
> |          9          |           72           |      58.37      |
> +-----------------+--------------------+-----------------+
>
>
> "A 3-layer LSTM has been used for the experiments  - how was this choice made?  What is the impact of this algorithm if the net was a 2-layer net as is typical in most large-scale LMs?"
>
> For the same number of parameters, we found 3 layered LSTMs to have better performance as compared to the 2-layered ones. This difference was not alleviated by hyperparameter tuning though it was not entirely extensive due to computational resources.
>
> "Table 3 is interesting to see how the cache model helps with rare words  and as such has applications in keyword spotting tasks. Were the hyper parameters of the cache tuned to perform better on rare words?  More details on the design of the cache model would have been useful."
>
> Analogous to the best model, the cache model was tuned to provide lower validation perplexity. The resulting efficacy on rare words is hence incidental but not entirely surprising.
>
> "You state that the gains obtained using the cache model were far less than what was obtained in Graves et al 2016 - what do you attribute this to?"
>
> We hypothesize that the reduction is due to the fact that our base language models have improved. Language models typically do well on common words while cache models are useful for rare words and those relating to past context. Graves et. al. does not use tied weights, for example, where tied weights were also shown to benefit rare words. As the language models get better at rarer words, or at using context, the usefulness of cache models diminishes given that there is no avenue left for improvement.

---

### Decision · Program_Chairs · 2018-01-29
**ICLR 2018 Conference Acceptance Decision**

**Decision:**

Accept (Poster)

**Comment:**

This paper presents a simple yet effective method for weight dropping for an LSTM that requires no modification of an RNN cell's formulation.  Experimental results shows good perplexity results on benchmarks compared to many baselines.  All reviewers agree that the paper will bring good contribution to the conference.